# Interconnected River–Lake Project Decreased $CO_2$ and $CH_4$ Emission from Urban Rivers

Chunlin Wang, Yuhan Xv, Siyue Li  and Xing Li *

Institute of Changjiang Water Environment and Ecological Security, School of Environmental Ecology and Biological Engineering, Key Laboratory for Green Chemical Process of Ministry of Education, Engineering Research Center of Phosphorus Resources Development and Utilization of Ministry of Education, Hubei Key Laboratory of Novel Reactor and Green Chemical Technology, Wuhan Institute of Technology, Wuhan 430205, China
* Correspondence: 18079102@wit.edu.cn

**Abstract:** Urban riverine networks are hotspots of $CO_2$ and $CH_4$ emissions, due to river impoundment and pollution. The river–lake connection is considered to be an important way to improve the ecological environment of urban rivers; however, its impact on $CO_2$ and $CH_4$ emissions from urban rivers and regulatory mechanisms are still unclear. Rivers and lakes have been studied separately by lots of traditional studies. In this study, we investigated the concentration and emission of $CO_2$ and $CH_4$ from March 2021 to December 2021 in an interconnected river–lake system in Central China. We found that the urban river–lake system was a hotspot of $CO_2$ and $CH_4$ emissions. $CO_2$ and $CH_4$ emissions from urban rivers were much higher than those from the lakes, which are 2.7 times and 11.9 times that of lakes, respectively. The correlation analysis indicated that the spatial variation of $CO_2$ and $CH_4$ emissions was determined by nutrient content. The abundant nutrients promoted microbial growth and consumed dissolved oxygen (DO), thus resulting in high emissions of $CO_2$ and $CH_4$ in the isolated urban rivers (UR). The average $CO_2$ and $CH_4$ emissions of urban rivers are 991.56 and 14.82 mmol m$^{-2}$ d$^{-1}$, respectively. The river–lake connection decreased the nutrients of urban rivers connected to lakes (LUR). The moderate nutrients wreaked in situ respiration, exhibiting moderate $CO_2$ emission in the LUR. The average $CO_2$ emission of LUR is 543.49 mmol m$^{-2}$ d$^{-1}$. The river–lake connection increased the DO concentrations in the LUR, inhibited methanogenesis, and enhanced $CH_4$ oxidation, reducing $CH_4$ emission from LUR sharply. The average $CH_4$ emission of LUR is 1.26 mmol m$^{-2}$ d$^{-1}$. A correlation analysis showed that the seasonal variations of $CO_2$ and $CH_4$ emissions were controlled by DO and T. Hence, the highest emissions of $CO_2$ were observed in the spring and the lowest in the winter, and the $CO_2$ emissions in spring were 10.7 times that in winter. The highest emissions of $CH_4$ were observed in the summer and the lowest in the winter, and the $CH_4$ emissions in summer were 6.6 times those in winter. The connection of urban rivers and lakes changes the environmental factors, thereby varying the production and emission of greenhouse gases. This study advanced the knowledge of the greenhouse gas emission response to the river–lake connection, providing the theoretical basis for greenhouse gas emission reduction from urban rivers.

**Keywords:** urban riverine networks; carbon dioxide; methane; carbon isotopes; urban river–lake system

## 1. Introduction

Carbon dioxide ($CO_2$) and methane ($CH_4$) are two important greenhouse gases. Recent monitoring by the World Meteorological Organization shows that the concentrations of $CH_4$ and $CO_2$ in the global atmosphere have reached $1869 \pm 2$ parts per billion and $408.0 \pm 0.1$ parts per million, a steady increase of 259% and 147%, respectively, since the Industrial Revolution [1]. Inland waters, including rivers, reservoirs, and lakes, are essential to the global carbon cycle and are a significant source of atmospheric greenhouse gases [2–4]. Global inland waters absorb about 5.1 PgC yr$^{-1}$ from the global terrestrial landscape,

and at the same time, they emit about 3.9 PgC yr$^{-1}$ of carbon dioxide [5] and 0.13 PgC yr$^{-1}$ of methane ($CH_4$) [6]. Annual $CO_2$ emissions from lakes and reservoirs are about 244–503 Tg $CO_2$ [7], accounting for about 21–29% of global $CO_2$ emissions from rivers and streams [8]. $CH_4$ emissions from rivers and streams were as high as 26.8 Tg $CH_4$ yr$^{-1}$ [6]. It is estimated that about 54.1 ± 41.0 TgC enters the atmosphere from $CO_2$ from rivers in China, 6.8 ± 8.8 TgC from lakes, and 1.1 ± 1.6 TgC from reservoirs. Rivers account for 87.3% of China's inland water $CO_2$ emissions [9]. China launched the carbon neutrality goal in 2020, promising to achieve carbon neutrality by 2060. Therefore, rivers, which account for the majority of carbon emissions, are becoming the focus of carbon emissions research.

With the rapid development of urbanization, urban water networks are increasingly affected by human activities. A large number of studies have found that the higher the level of urbanization, the more greenhouse gas (GHGs) emissions of urban water networks are exceptionally high, at even several times or even dozens of times that of other types of rivers [10–14]. In the Chaohu Lake Basin, the greenhouse gas emissions of different landscape types were studied, and it was found that the $CO_2$ and $CH_4$ of urban rivers were two times and seven times that of other varieties [12]. Some other studies found that GHG emissions increased significantly once having flowed through the urban area [15,16]. These studies have suggested that the magnitude and spatiotemporal variations of GHG emissions are affected by weather conditions (e.g., water temperature and wind speed), nutrient level (e.g., nitrogen, phosphorus, and carbon), hydrological conditions (e.g., flow velocity and water depth) and biological activities [17,18]. Impacts by human activities include increased pollution levels of urban water networks, significant inputs of organic matter, and changes in hydrogeomorphology [19]. For example, the water level of urban rivers is low due to water shortage, and the diffusion path of $CH_4$ is short, which can reduce the oxidation of $CH_4$ and increase emissions [20]. Moreover, urban rivers are shallow, and the surface runoff formed by rainfall will bring sediment into urban rivers, causing the sediment layer of urban rivers to gradually thicken, which will promote $CO_2$ and $CH_4$ emissions [21,22]. Low water depth of urban rivers also results in low water hydrostatic pressure, leading to GHG emissions through ebullition [23]. A study of urban rivers in Shanghai found that the escape of $CH_4$ through ebullition has no time change and accounts for 99% of $CH_4$ emissions [24]. A large amount of nutrient input will cause microorganisms to decompose organic matter to produce a large amount of $CO_2$ and, at the same time, consume oxygen to form an anaerobic environment in urban rivers [25], creating favorable conditions for methanogens to produce $CH_4$. However, the enrichment of nutrients in rivers and lakes will also promote the growth of environmental algae, thereby increasing primary production, taking $CO_2$ in rivers and lakes, and reducing $CO_2$ emissions from rivers and lakes [26–28].

Shallow lakes are inland water that contributes GHG emissions due to their large area and eutrophication [29,30]. Compared with deep lakes, shallow lakes are more vulnerable to eutrophication due to the low buffer capacity for nutrient input [31]. A number of studies reported that eutrophic status plays a critical role in impacting the magnitude of GHG emissions from lakes [7,32,33]. It was found that eutrophication can both increase GHG emissions due to stimulating mineralization and decrease GHG emissions due to the enhancement of primary production [34]. However, Zhou et al. (2023) reviewed that $CO_2$ emission from lakes (mean ± SD:20.0 ± 26.0 mmol m$^{-2}$ d$^{-1}$) is notably lower than those from rivers (mean ± SD:179.7 ± 149.5 mmol m$^{-2}$ d$^{-1}$) of China due to urbanization [9]. Zheng et al. (2022) reported that the mean value of $CH_4$ emission from lakes is about half of that from rivers globally [35]. The reason for the relatively lower GHG emissions from lakes is the lower nutrient levels in lakes. Lakes connecting to rivers could play the role of converter and water accumulator [36]. Thus, the river–lake connection is considered to be an important way to improve the ecological environment of urban rivers [37]. Since 2010, more than 100 cities in China have constructed interconnected river–lake projects. However, lakes and rivers have traditionally been studied as separate systems for GHG emissions, and a knowledge gap still exists in the understanding of spatiotemporal variations of GHG

emissions in an urban interconnected river–lake system. This study was conducted in an interconnected river–lake system in Wuhan, Central China. The objectives were (1) to estimate the spatial and seasonal variations and correlated environmental factors associated with the emissions of $CO_2$ and $CH_4$, and (2) to reveal the impact of river–lake connection on $CO_2$ and $CH_4$ emission in urban rivers.

## 2. Materials and Methods

### 2.1. Study Area

The urban river–lake system is located in Wuhan City of Hubei Province, Central China (Figure 1). Wuhan is located in the middle reaches of the Yangtze River, the longest river in China. The study area is characterized by a typical subtropical monsoon climate, and the annual average temperature is 15.8–17.5 °C. The annual precipitation is 1150–1450 mm, and the rainfall is mainly from April to September, accounting for about 70% of the annual rainfall. The complex river–lake system in Wuhan, consisting of 166 lakes and 2 big rivers (Yangtze River and Han River), accounts for a quarter of the total urban area. Lake Tangxun is the largest urban lake in the world, with an area of 47.6 km$^2$. This study area is an area with abundant surface water resources. However, with extensive economic development and population growth, river and lakes in Wuhan suffered heavy pollution [38].

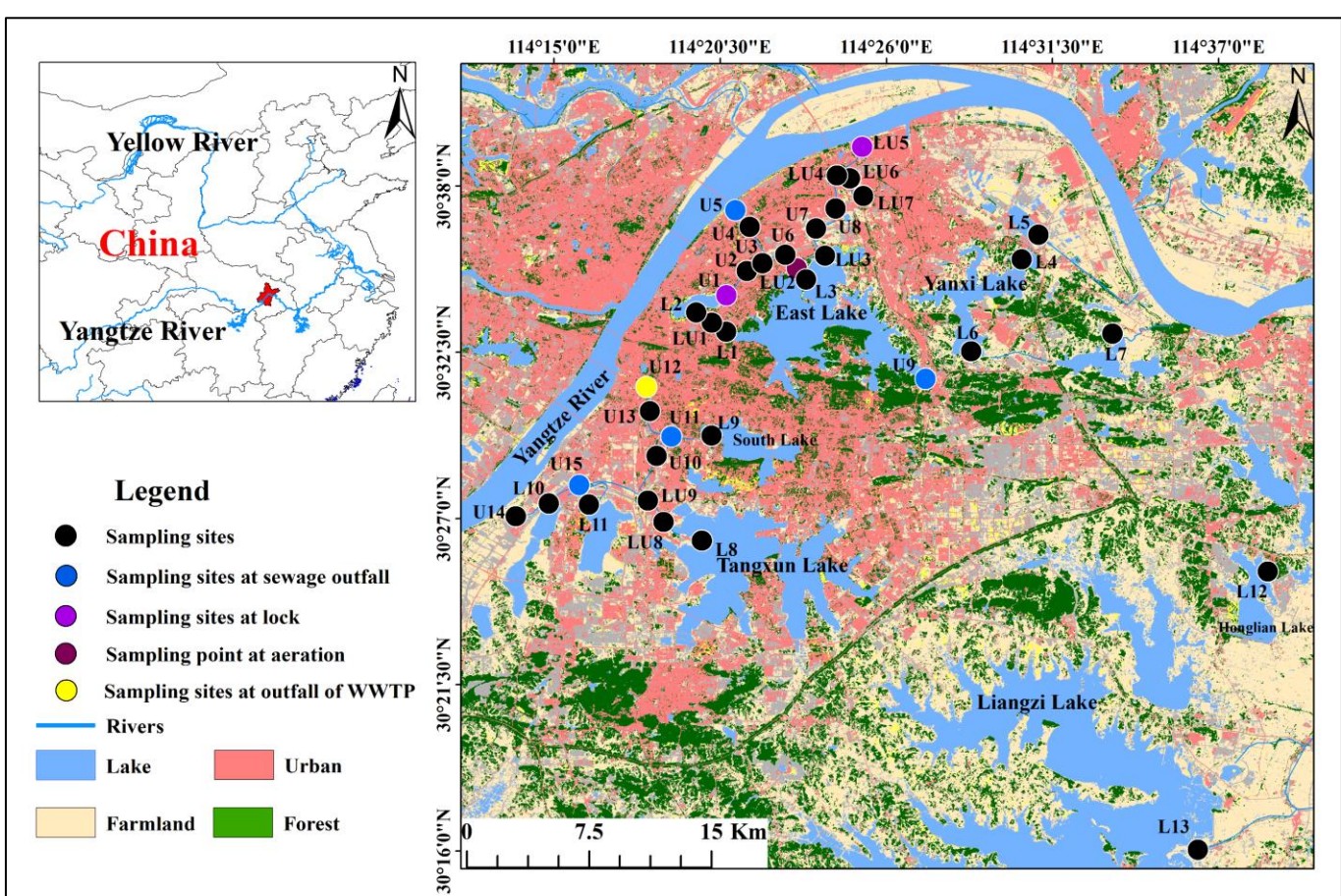

**Figure 1.** Geographic locations and sampling sites of the interconnection river–lake system in Wuhan City, Central China. U1-U15 are UR sites, LU1-LU9 are LUR sites, and L1-L13 are lake sites.

There are many lakes distributed in the urban area on the south bank of the Yangtze River, and the major source of water to these lakes is precipitation, groundwater recharge, and domestic sewage. To improve the water quality and quantity of urban rivers, the rivers and lakes are connected by channels, thereby forming a complex plain urban water system. However, although the rivers in the city are physically connected with the lakes

by channels, the actual connectivity between the rivers and lakes is quite different due to water level, sluice, and dams. Some urban rivers are still isolated from the lakes, so the urban water system was divided into isolated urban rivers (UR), urban rivers connected to lakes (LUR), and lakes.

### 2.2. Experimental Samples and Analysis Methods

Samples were collected every three months from 2021 to 2022. A total of 37 sampling points were set up, and 13, 15, and 9 samples were set for lakes, UR, and LUR, respectively. The water temperature (T), dissolved oxygen (DO), pH, and electrical conductivity (EC) were measured with a portable water quality parameter sensor (Hach Company, CO Loveland, USA). Water samples of 200 mL were filtered using 0.45 μm cellulose acetate membranes to measure the dissolved inorganic nitrogen (TDN) and dissolved organic carbon (DOC). A total of 50 mL of water sample was filtered through a needle filter with a 0.45 μm pore size, and then 1–2 drops of 1 mol $L^{-1}$ hydrochloric acid were added to fix the $NH_4^+$. $NH_4^+$-N, $NO_3^-$-N, and $NO_2^-$-N were determined by a flow analyzer (San++, Skalar, Netherlands). TDN and DOC were determined using a total organic carbon/total nitrogen analyzer (multi N/C 2100, Jena, Germany). A depth of surface water of 0–20 cm was collected from rivers and lakes. A 100 mL syringe pre-loaded with saturated $HgCl_2$ was used to collect 50 mL of water samples to make a 50 mL headspace, and the gas in the headspace was transferred to the airbag after $CO_2$ and $CH_4$ were in the water at equilibrium with the air. The gas sample in the airbag was measured within 24 h. The concentration of $CO_2$ and $CH_4$ in the equilibrated gas was measured using gas chromatograph (7890 B, Agilent Technologies, Santa Clara, CA, USA), and the instrument detection limits were 0.1 ppm. An amount of 100 mL surface water was collected for chlorophyll a (Chl-a); Chl-a was extracted from the water samples using a 90% buffered acetone solution, and the concentration was determined using an ultraviolet (UV) spectrophotometer (Shimadzu, UV-2600 PC). We collected dissolved inorganic carbon (DIC) samples using 250 mL bottles and then added 1–2 drops of saturated $HgCl_2$ solution. $\delta^{13}$C-DIC values were analyzed by a GasBench II System interfaced to a Delta V Plus IRMS (Thermo Scientific, Bremen, Germany). The method detection limit and long-term standard deviation for $\delta^{13}$C-DIC as $CO_2$ were approximately 150 nmol and 0.1‰.

### 2.3. Experimental Methodology

The surface water $CO_2$ and $CH_4$ concentrations in rivers and lakes ($C_{sample}$) were calculated using Henry's law [39] to obtain the $CO_2$ and $CH_4$ concentrations in the samples. Calculate the $CO_2$ and $CH_4$ concentrations (μmol $L^{-1}$) by Formulas (1)~(4).

$$K_H^{CC} = C_a/C_g \tag{1}$$

$C_a$ is the concentration of $CO_2$ and $CH_4$ in the liquid phase in the headspace equilibrium state (μmol $L^{-1}$), $C_g$ is the concentration of $CO_2$ and $CH_4$ in the gas phase (μmol $L^{-1}$), and $K_H^{CC}$ is a constant. $K_H^{CC}$ is unknown in this formula, and we need the following procedure:

$$TK_H = 12.2K_H^{CC} \tag{2}$$

$K_H$ is the Henry coefficient, and $T$ is the temperature when $T = 298.15$ K:

$$K_H = 0.0409K_H^{CC} \tag{3}$$

when $T = 298.15$ K, the $K_H$ values of $CO_2$ and $CH_4$ are, respectively, $3.6 \times 10^{-2}$ and $1.4 \times 10^{-3}$ [40,41], from which Ca be calculated, and then the following formula is used:

$$(C_a \times V_a + C_g \times V_g) \div V_{sample} = C_{sample} \tag{4}$$

$V_a$ is the liquid volume (L) in the headspace equilibrium state, $V_g$ is the gas volume (L) in the headspace equilibrium state, $V_{sample}$ is the sample volume before equilibrium (L), and $C_{sample}$ is the concentration of $CO_2$ and $CH_4$ in the sample (μmol $L^{-1}$).

When the $CO_2$ and $CH_4$ concentrations in rivers and lakes are calculated, we also need to calculate the flux (*F*), which is calculated using the water–air interface diffusion model [42], and the calculation formula is as follows:

$$F = K \times \left( C_w - C_{eq} \right) \tag{5}$$

*K* is the gas transfer velocity at the water–air interface, and $C_w$ is the concentration of $CO_2$ and $CH_4$ in the surface water of rivers and lakes (μmol $L^{-1}$). $C_{eq}$ is the concentration of $CO_2$ and $CH_4$ in the river and lake at the sampling point when the concentration of $CO_2$ and $CH_4$ in the surface water of the river and lake reaches the equilibrium with the attention in the atmosphere (μmol $L^{-1}$). In this study, *K* is calculated using the calculation method of Wanninkhof et al. [43]. The formula is as follows:

$$K = K_{600} \times \left( \frac{Sc}{600} \right)^{-n} \tag{6}$$

$$Sc_{CO_2} = 1911.1 - 118.11T + 3.527T^2 - 0.041320T^3 \tag{7}$$

$$Sc_{CH_4} = 1897.8 - 114.28T + 3.2902T^2 - 0.003906T^3 \tag{8}$$

*n* is the coefficient and takes 0.5, *Sc* is the Schmidt number of the gas, *T* is the temperature of the sampling point, and $K_{600}$ is the diffusion rate of the gas when *Sc* is equal to 600. $K_{600}$ is controlled by physical factors related to water flow and is calculated from models of water flow and wind speed [44–46]. In this study, we choose the wind speed model [46], and the calculation formula is as follows:

$$K_{600} = 1.91 \times e^{0.35 \times u_{10}} \tag{9}$$

$u_{10}$ is the average wind speed at 10 m from the sampling point (m $s^{-1}$). The wind speed at the sampling point adopts the average wind speed reported by the Meteorological Bureau on the day. The nonparametric Kruskal–Wallis method (K-W test) was used to test whether there were statistical differences in the spatial and temporal $CO_2$ and $CH_4$ concentrations in rivers and lakes. Kolmogorov–Smirnov was used to test the normality of environmental factors, gas concentration, and flux, and the test results did not conform to normality. Therefore, the Spearman correlation was used to analyze the correlation level between $CO_2$ and $CH_4$ fluxes and environmental factors, using IBM SPSS 26.0 and R 4.1.0 software. The spatial and temporal distribution characteristics of $CO_2$ and $CH_4$ concentrations and fluxes were drawn using Origin 2021 software.

## 3. Results

### 3.1. Spatiotemporal Variation of Environmental Factors

The interconnected river–lake system exhibited significant spatial and temporal variations of environmental factors (Table 1). Except for water temperature, all other environmental factors have significant differences among the rivers and lakes. The mean value of DO (9.4 ± 1.8 mg $L^{-1}$) concentration and pH (8.2 ± 0.5) was similar to the lakes, but was much higher than those of values in the UR. The mean value of EC (361.8 ± 54.0 μs $cm^{-1}$) and TN (2.1 ± 1.3 mg $L^{-1}$) was similar to the lakes, but was much lower than that in the UR. The mean concentration of DOC and Chl-a in the LUR was lower than that in the UR and lakes. The abundances of bacteria in the LUR were the lowest, and those values in the UR were the highest. Except for the pH values and TDN, there were significant differences in the mean value of T, DO, EC, Chl-a, and DOC. The DO concentrations were higher in the spring and summer than those values in the autumn and winter. The mean

value of EC was the highest in the winter and the lowest in the summer. The highest concentrations of Chl-a and DOC were observed in the autumn, and the lowest were observed in the winter. The highest abundances of bacteria were observed in the spring, with no significant differences in the other seasons. Because of the hydrological connection, the hydrodynamic conditions of the connected river channels are improved, which can promote the dissolution of oxygen in the air in the water body and enable DO to enter the deep water columns, thereby increasing the DO of the river. It will also cause the flow velocity of connected rivers to increase and the carbon residence time to decrease, which is not conducive to algae reproduction; the carbon dioxide produced will also decrease, and the pH will increase accordingly.

**Table 1.** Statistical characteristics of riverine and lake physicochemical indicators.

| Factors | Unit | Type | | | Season | | | |
|---|---|---|---|---|---|---|---|---|
| | | UR | LUR | Lakes | Spring | Summer | Autumn | Winter |
| pH | - | $7.8 \pm 0.6$ [a] | $8.2 \pm 0.5$ [b] | $8.3 \pm 0.6$ [b] | $8.0 \pm 0.5$ [a] | $8.0 \pm 0.8$ [a] | $8.4 \pm 0.6$ [b] | $7.9 \pm 0.5$ [a] |
| T | °C | $20.6 \pm 7.2$ [a] | $19.5 \pm 7.5$ [a] | $20.1 \pm 8.7$ [a] | $19.3 \pm 1.4$ [a] | $30.1 \pm 1.9$ [b] | $20.7 \pm 1.3$ [a] | $9.8 \pm 1.8$ [c] |
| EC | μs/cm | $516.5 \pm 87.9$ [a] | $361.8 \pm 54.0$ [b] | $348.5 \pm 97.4$ [b] | $426.2 \pm 74.2$ [a] | $395.2 \pm 109.7$ [b] | $397.3 \pm 121.6$ [b] | $465.6 \pm 123.5$ [c] |
| Chl a | μg/L | $38.2 \pm 77.5$ [a] | $21.9 \pm 26.4$ [b] | $46.5 \pm 47.4$ [a] | - | $41.6 \pm 54.6$ [a] | $42.7 \pm 74.0$ [ab] | $27.2 \pm 43.4$ [b] |
| DO | mg/L | $5.9 \pm 4.2$ [a] | $9.4 \pm 1.8$ [b] | $9.9 \pm 2.7$ [b] | $7.8 \pm 3.2$ [a] | $7.1 \pm 4.1$ [a] | $8.7 \pm 4.0$ [b] | $8.8 \pm 3.0$ [b] |
| TDN | | $10.4 \pm 4.7$ [a] | $2.1 \pm 1.3$ [b] | $1.6 \pm 1.0$ [b] | $3.9 \pm 3.4$ [a] | $4.6 \pm 5.0$ [a] | $4.1 \pm 4.4$ [a] | $5.0 \pm 5.9$ [a] |
| DOC | | $6.9 \pm 2.9$ [a] | $4.8 \pm 2.7$ [b] | $7.0 \pm 3.9$ [a] | $4.7 \pm 1.5$ [ac] | $5.8 \pm 3.5$ [a] | $9.8 \pm 2.5$ [b] | $4.4 \pm 1.2$ [c] |
| 16 S | $10^{11}$ Copies/L | $2.8 \pm 3.0$ [a] | $1.37 \pm 0.9$ [b] | $1.8 \pm 1.0$ [c] | $4.0 \pm 4.0$ [a] | $2.1 \pm 2.0$ [b] | $1.8 \pm 1.5$ [b] | $1.5 \pm 1.1$ [b] |
| $\delta^{13}$ C-DIC | ‰ | $-10.7 \pm 1.7$ [a] | $-9.2 \pm 2.6$ [b] | $-7.6 \pm 2.8$ [c] | - | $-10.9 \pm 1.1$ [a] | $-9.1 \pm 2.0$ [b] | $-7.8 \pm 3.4$ [b] |

Notes: All data are shown as the mean annual value $\pm$ standard errors of the means. Note that the affiliated letters [a], [b], and [c] indicate statistical differences at the 95% confidence level among rivers and lakes ($p < 0.05$, Kruskal–Wallis test).

### 3.2. Spatiotemporal Variation of $CO_2$ Concentrations and Emission Fluxes

The $CO_2$ concentrations ranged from 55.9 to 2246.1 μmol $L^{-1}$, with an overall mean value of $539.1 \pm 512.5$ μmol $L^{-1}$ (Figure 2). All these values were higher than those of the atmosphere, resulting in the urban river–lake system being the emission source of $CO_2$ in the atmosphere. The concentrations of dissolved $CO_2$ varied spatially and temporally. The $CO_2$ concentration was highest in the spring, followed by the summer, and the lowest in the autumn and winter without significant differences in the UR. In the LUR and lakes, mean values of $CO_2$ concentration were higher in the spring, and there was no difference among the other seasons. Spatially, concentrations of $CO_2$ in the LUR were lower than those in the UR, and higher than those in the lakes.

The emission fluxes of $CO_2$ ranged from 48.2 to 4351.0 mmol $m^{-2}$ $d^{-1}$, with a mean value of $614.7 \pm 727.4$ mmol $m^{-2}$ $d^{-1}$. In the UR, the highest $CO_2$ emissions were observed in the summer, followed by the spring, which is contrary to the seasonal dynamics trend in $CO_2$ concentrations. However, the seasonal variation pattern was consistent with that of $CO_2$ concentrations; that is, the highest $CO_2$ emission was observed in the spring and the lowest in the winter. In addition, $CO_2$ emissions in the LUR were lower than those in the UR, and higher than those in the lakes, which was consistent with the dynamics pattern of $CO_2$ concentrations. The spatial difference of $CO_2$ is mainly caused by the difference in nutrient load, which is greatly affected by the pollution emission from human activities. The main factors affecting the time difference of $CO_2$ are the temperature and the time change in algae, because the main way of $CO_2$ consumption in the study area is the primary production of algae.

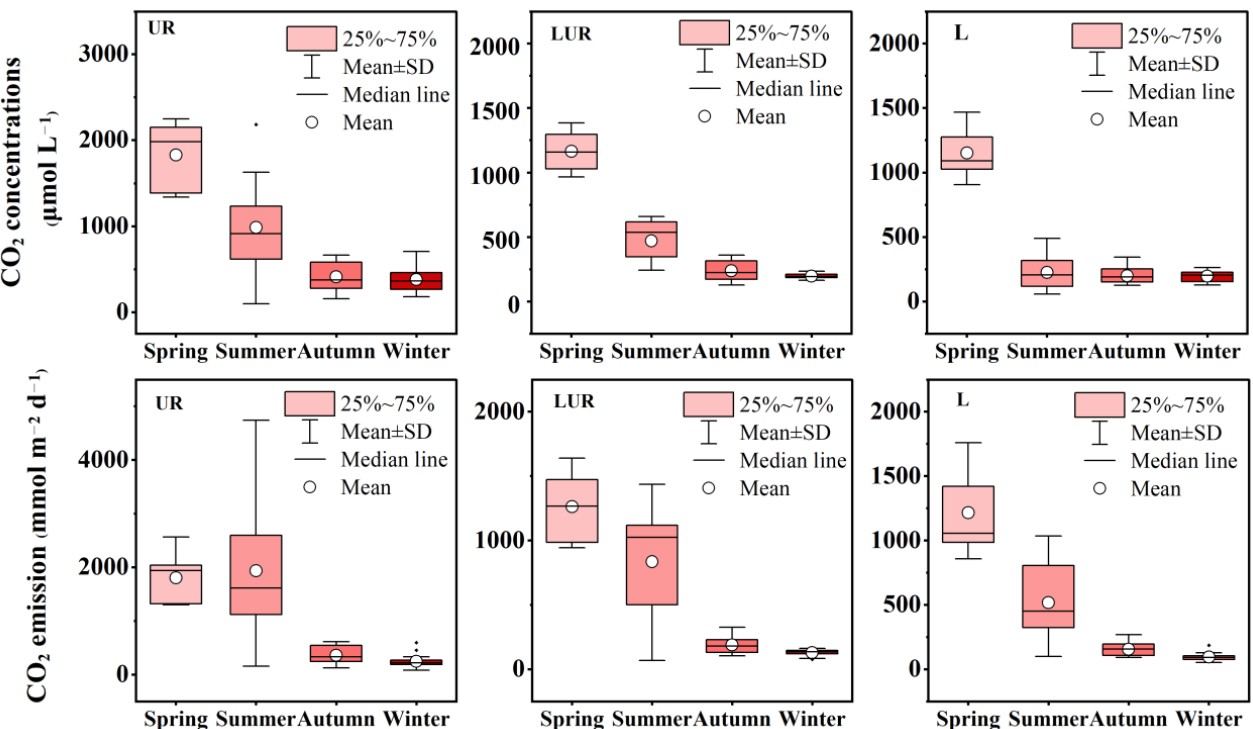

**Figure 2.** Seasonal and spatial variations of $CO_2$ concentrations and emission fluxes in the interconnection urban river–lake system.

### 3.3. Spatiotemporal Variability in $CH_4$ Concentrations and Fluxes

The temporal and spatial variation characteristics of $CH_4$ concentrations and emissions are presented in Figure 3. The $CH_4$ concentrations varied from 0.02 to 78.6 $\mu mol\ L^{-1}$, with a mean value of $5.5 \pm 11.5\ \mu mol\ L^{-1}$. All these concentrations indicated the saturation of $CH_4$, leading to the emission source of $CH_4$. The mean value of $CH_4$ concentrations was the highest in spring and the lowest in winter. All the values in the UR were much higher than those values in the LUR and lakes, while no significant difference existed between the LUR and lakes.

$CH_4$ emission ranged from 0.01 to 158.1 $\mu mol\ L^{-1}$, with a mean value of $6.37 \pm 17.2\ \mu mol\ L^{-1}$. According to the dynamics pattern of $CH_4$ concentrations, $CH_4$ emissions in the UR were the highest, while no significant difference existed between the LUR and lakes. Seasonal variations of $CH_4$ emissions were also estimated, with the highest values in the summer and the lowest values in the winter, exhibiting different dynamics patterns with $CH_4$ concentrations. DO and temperature mainly affect the temporal and spatial changes in $CH_4$. Anaerobic conditions are conducive to the production of $CH_4$ and inhibit the oxidation of $CH_4$. The level of temperature will affect the activity of methanogens and thus affect the production of $CH_4$.

### 3.4. $\delta 13$. C-DIC Signatures of Dissolved Inorganic Carbon

The $\delta 13$C-DIC values of dissolved inorganic carbon in the river–lake system are shown in Figure 4. The $\delta 13$C-DIC varied from $-13.88$ to $-2.37‰$, with a mean value of $-9.26 \pm 2.7‰$. The lower values of $\delta 13$C-DIC were observed, and in the UR, the higher values were observed in the lakes. The variation range of $\delta 13$C-DIC was relatively large in the LUR. The $\delta 13$C-DIC had a clear seasonal variation pattern in the LUR and lakes, that is, gradually increasing from summer to winter. In particular, although the lowest value of $\delta 13$C-DIC was still observed in the summer, there is no significant difference between autumn and winter. The $\delta 13$C-DIC values were close to the endmember of the atmosphere in the winter, and those values were close to the endmember value of organic matter degradation. Therefore, the low $\delta 13$C-DIC value in summer is due to a large

amount of decomposition of organic matter, and the reason for the high value in winter is that the decomposition intensity of organic matter decreases, the proportion of $CO_2$ produced decreases, and the atmospheric source increases, which promotes the increase in the δ13C-DIC value.

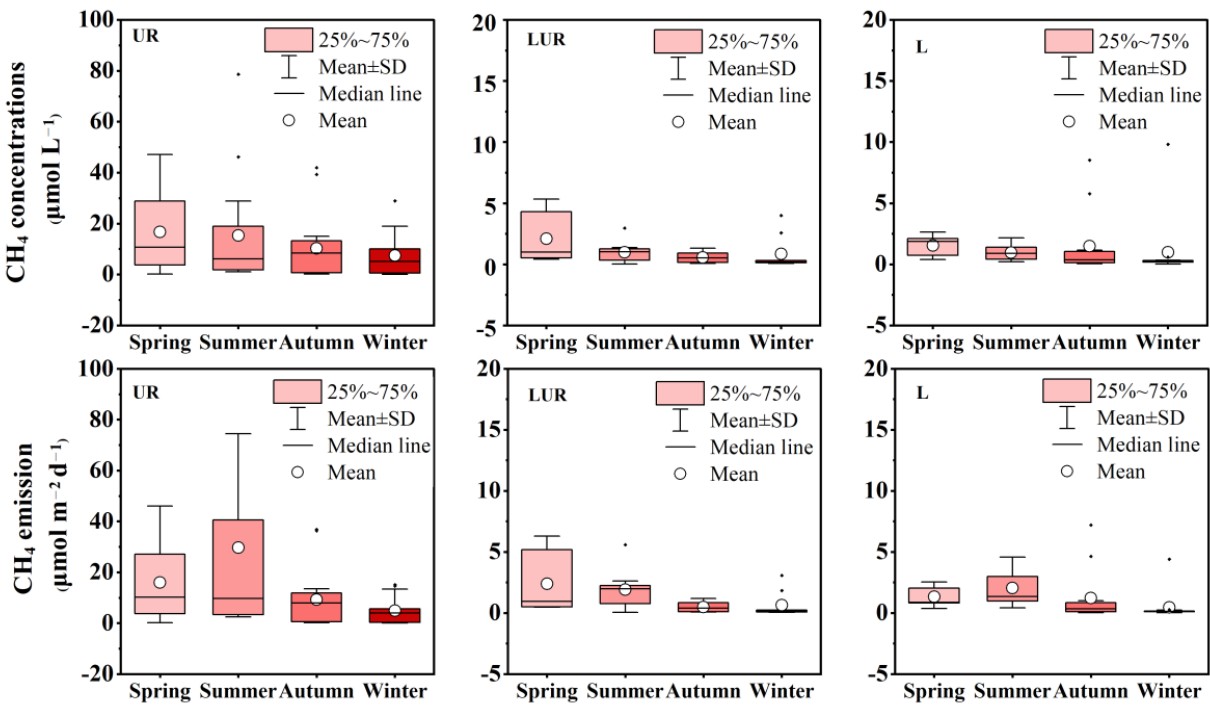

**Figure 3.** Seasonal and spatial variations of $CH_4$ concentrations and emission fluxes in the interconnection urban river–lake system.

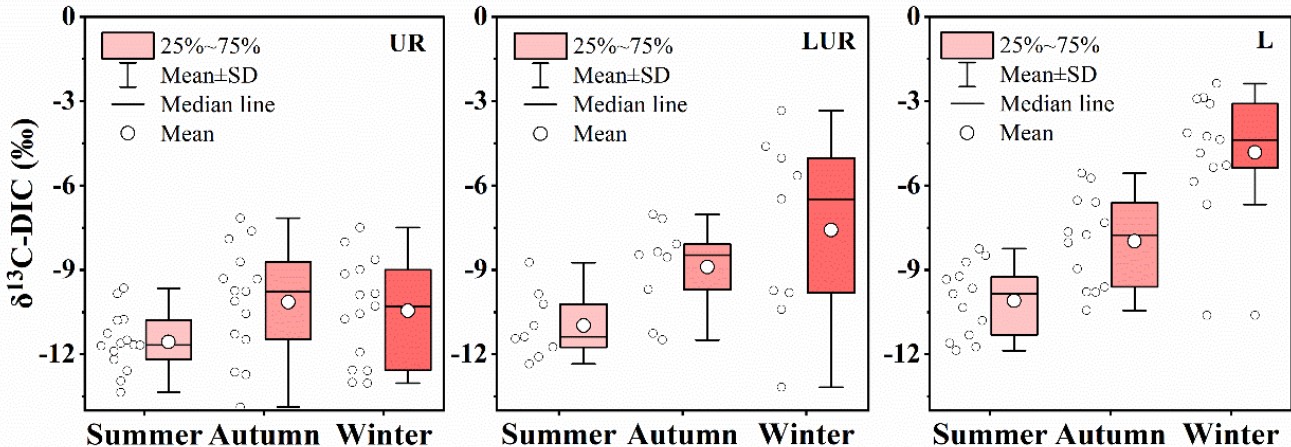

**Figure 4.** Seasonal and spatial variations of $\delta^{13}$C-DIC values in the interconnection urban river–lake system.

*3.5. Correlation between $CO_2$ and $CH_4$ Emissions and Environmental Factors*

The emissions of $CO_2$ and $CH_4$ are affected by various environmental factors. Spearman correlation analysis was used to estimate the correlation between $CO_2$ and $CH_4$ emissions and environmental factors (Figure 5). The Spearman correlation analysis showed that $CO_2$ emission was positively correlated with EC and TDN ($p < 0.05$) and negatively correlated with DO, pH, and Chl-a in all seasons. $CO_2$ emission was positively correlated with DOC and abundance of 16S in the spring, but not significantly correlated with DOC in the other seasons. Spatially, T was positively correlated with $CO_2$ emission in all the rivers

and lakes. $CO_2$ emission was negative with DO and pH in the UR. There is no significant correlation observed between $CO_2$ emission and other factors in the LUR. DO concentration was negatively correlated with $CO_2$ emission in the lakes. The decomposition of organic matter to produce $CO_2$ requires the consumption of oxygen, and the more $CO_2$ is produced, the more oxygen is consumed. The primary production of algae consumes $CO_2$, so the higher the concentration of chlorophyll a, the greater the $CO_2$ consumption. Therefore, $CO_2$ emissions are negatively correlated with DO and Chl-a.

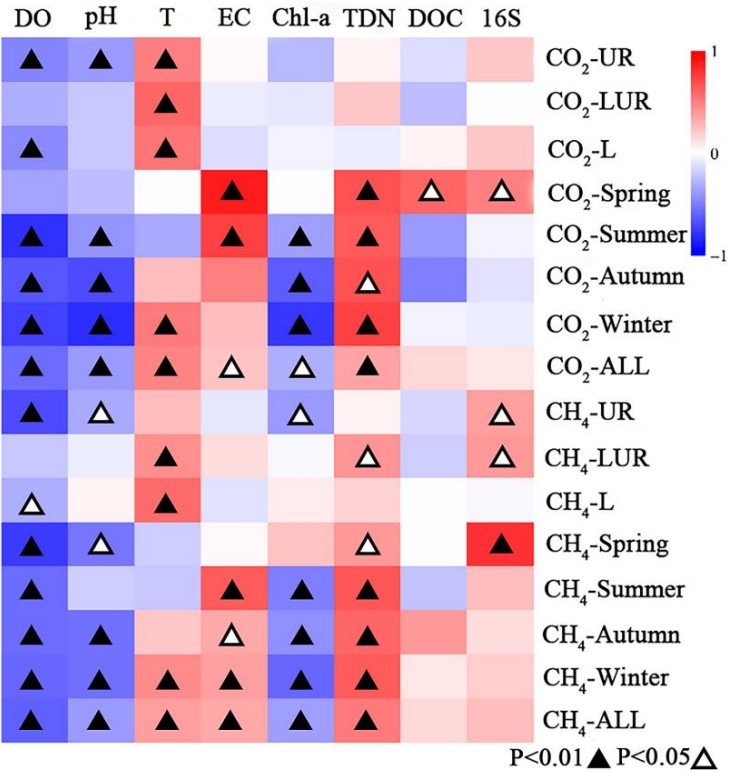

**Figure 5.** Heat map of Spearman correlation coefficients between $CO_2$ and $CH_4$ emission and environmental factors.

Chl-a and $\delta^{13}$C-DIC are the most important indicators that reveal the processes of $CO_2$ production and consumption. Therefore, a more detailed correlation analysis between these factors and $CO_2$ emission was performed in this study (Figures 6 and 7). For the UR, $CO_2$ emission was negatively correlated with Chl-a in the autumn. For the LUR, $CO_2$ emission was negatively correlated with Chl-a in the winter. For the lakes, $CO_2$ emission was negatively correlated with Chl-a in the winter, and positively correlated in the summer. The most significant correlation between $CO_2$ emission and Chl-a was observed in the lakes. Dissolved $CO_2$ was negatively correlated with $\delta^{13}$C-DIC in all the seasons, rivers, and lakes (Figure 7). The $CO_2$ produced by the decomposition of organic matter decreases in winter, and then the primary production of algae consumes $CO_2$, and the impact on $CO_2$ emissions is amplified. Therefore, winter $CO_2$ emissions had the strongest correlation with Ch-a. Due to the high nutrient load, the $CO_2$ produced by the decomposition of organic matter in UR is much higher than that in LUR and lakes, so algae consume $CO_2$ and have less impact on UR's $CO_2$ emissions than in LUR and lakes.

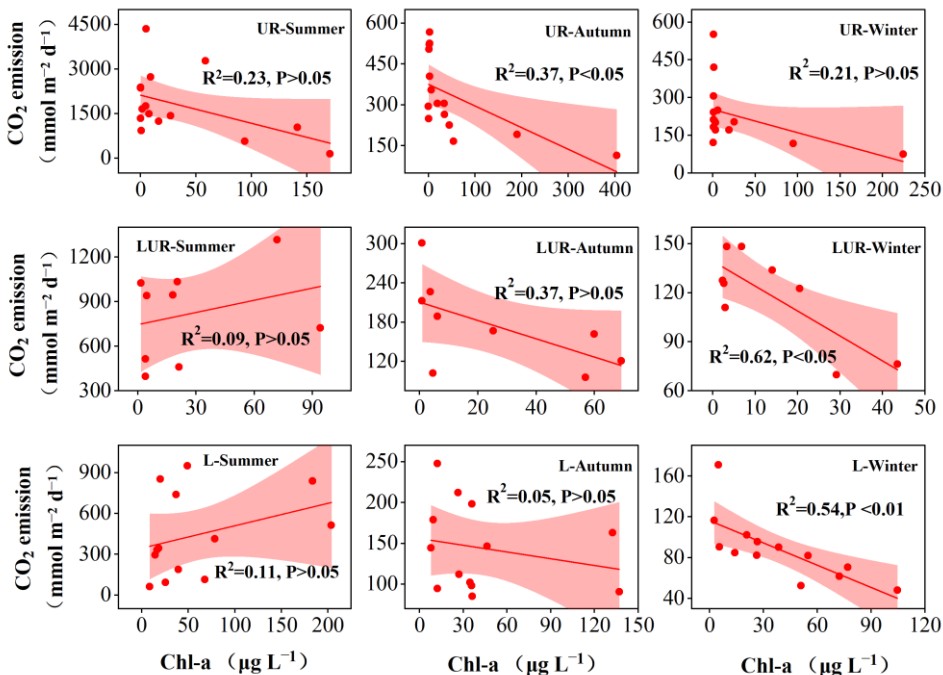

**Figure 6.** Seasonal and spatial variations of correlation between $CO_2$ emission and Chl-a.

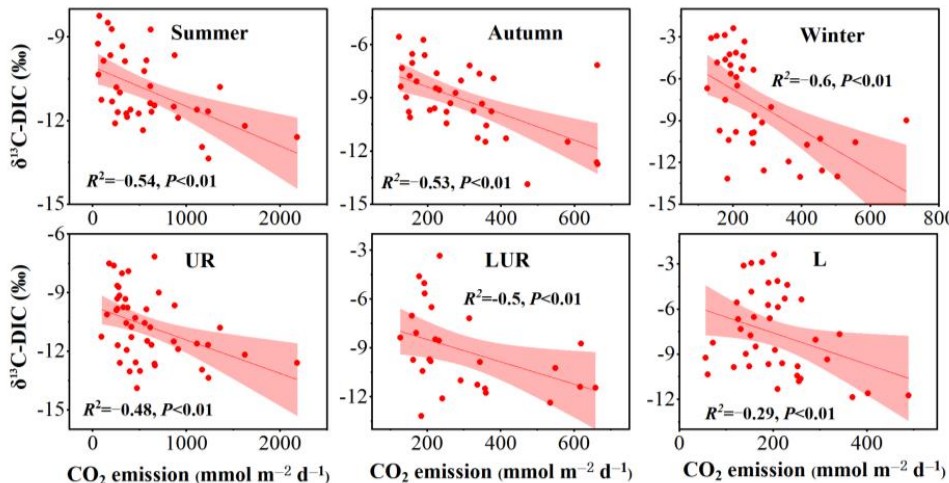

**Figure 7.** Seasonal and spatial variations of correlation between $CO_2$ emission and $\delta^{13}C$-DIC.

The Spearman correlation analysis showed that $CH_4$ emission was positively correlated with T, EC, and TDN, and negatively correlated with DO, pH, and Chl-a among all the data (Figure 5). The correlation between $CH_4$ emission and environmental factors in different seasons was consistent with all the data. Spatially, a negative correlation was observed between $CH_4$ emission and DO, pH, and Chl-a, but no significant correlation was observed with other factors in the UR. TN and T were positively correlated with $CH_4$ emission in the UR. $CH_4$ emission was correlated with T and DO in the lakes. $CH_4$ is produced under anaerobic conditions, because the production of $CO_2$ consumes oxygen, reduces the concentration of DO in the river, and creates favorable conditions for the production of more $CH_4$, so there is a negative correlation. Chl-a is also negatively correlated with $CH_4$, because the more algae there are, the more oxygen will be produced in the surface water due to primary production, and the high DO concentration in the surface water will increase the oxidation of $CH_4$ in the emission process.

## 4. Discussion

### 4.1. Effect of River–Lake Connection on Environmental Factors

Hydrological connectivity could affect the diffusion and distribution of environmental factors in inland waters [47]. Environmental factors exhibited significant differences between the UR and LUR, suggesting that the factors are affected by the river–lake connection. The Kruskal–Wallis test showed that DO concentrations were much higher in the LUR than those in the UR, which is consistent with previous studies [48–50]. Yue et al. (2015) found that the DO concentration in connected wetlands was higher than that in isolated wetlands. The vertical mixing is uniform with good hydrological connectivity, which is conducive to the transfer of DO from surface water to deep water columns, resulting in higher DO in the rivers. However, a water body with poor hydrodynamic conditions is more prone to vertical stratification, which inhibits the vertical transfer of DO and leads to the hypoxia conditions of the rivers [51]. Stratification of rivers is conducive to the sedimentation of suspended particles, causing the accumulation of nutrients such as carbon and nitrogen in disconnected rivers and lakes [52]. Therefore, TDN and DOC concentrations were significantly lower in the LUR than those in the UR. The river–lake connection increased the pH values in the LUR, due to the absence of production of large amounts of $CO_2$. EC values reflect the total content of anions and cations in the water body [53]. The EC values of the LUR were influenced by the connection of rivers and lakes, which was very close to the EC values of lakes, but much lower than those values in the UR. In addition to the above physical factors, Chl-a and the abundance of 16S are also affected by the river–lake connection. High flow velocities and the short residence time of carbon under good hydrological connectivity are not conducive to the growth of algae and bacteria [54]. Thus, the concentration of Chl-a and 16S are supposed to be higher in lakes and reservoirs than those in rivers. In this study, the concentration of Chl-a in the isolated urban river (UR) had a similar level with the lakes, due to the poor hydrological connectivity of the UR. The relative lower concentrations of Chl-a were observed in the LUR (Table 1).

### 4.2. Control Processes on $CO_2$ and $CH_4$ Emissions

$CO_2$ emissions from river and lake surfaces are mainly controlled by the balance of respiration and primary production [55,56]. Previous studies suggested that external biodegradation from terrestrial organic carbon and in situ biodegradation of organic carbon are the main sources of aqueous $CO_2$ [54]. However, the contribution of external biodegradation-sourced $CO_2$ is limited in urban rivers, while the in situ biodegradation of organic carbon is likely more relevant to aqueous $CO_2$ [57]. This is supported by the significant negative correlation between $CO_2$ emission and DO. In addition, the inflow of treated and untreated sewage has been considered a source that cannot be ignored in urban rivers [14,31]. $CO_2$ concentrations in the final effluent of water treatment plants (WWTPs) can reach ~30 mmol L$^{-1}$ [58]. This is supported by the mean value of $\delta^{13}$C-DIC in the UR ($-10.7 \pm 1.7$) and LUR ($-9.2 \pm 2.6$‰), which were very close to that found in WWTP effluent in Brazil ($-12.2$‰) [59] and in China ($-12$‰) [60]. The absorption of $CO_2$ by algae photosynthesis is the main process of $CO_2$ consumption in rivers and lakes [32,56]. This is supported by the significant negative correlation between $CO_2$ emission and Chl-a. The mean Chl-a concentrations of this study were much higher than those in other rivers and lakes, and the significant negative correlation between $CO_2$ emission and Chl-a both indicated that the impact of primary production on $CO_2$ emission should be considered in this study [27,54,61]. Xu et al. (2018) reported that DIC in a river–lake continuum originated from $^{13}$C-depleted sources with a mean $\delta^{13}$C-DIC value of $-18.5$‰ [62]. However, the inflow of sewage and algae photosynthesis could be what enriched $\delta^{13}$C-DIC values in this study. $CH_4$ emission depends on anaerobic methanogenesis, aerobic methanotrophy, and $CH_4$ oxidation [63]. Methanogenesis is sensitive to DO concentrations, and high $CH_4$ emission always been observed in anaerobic rivers and lakes [64–66]. The lower DO associated with the higher $CH_4$ emission showed that methanogenesis-sourced $CH_4$ contributed to $CH_4$ emission in the UR. $CH_4$ oxidation is also sensitive to DO, and the significant negative

correlation between $CH_4$ emission and DO indicates that $CH_4$ oxidation dominated the decrease in the $CH_4$ emission in the LUR and lakes.

*4.3. Control Factors on $CO_2$ Emissions*

Environmental factors control the production and consumption process of $CO_2$ to regulate the spatiotemporal variations of $CO_2$ emission [67]. According to the correlation analysis, DO and T were the best predictors of $CO_2$ emission in the river–lake system, as suggested by the strongest correlation between $CO_2$ emission and DO and T (Figure 5). $CO_2$ is the end product of respiration, strongly affected by temperature [6,68]. Water temperature also plays a vital role in gas solubility and gas transfer velocity (k) at the water–air interface [46]. An increasing water temperature can decrease gas solubility and increase the value of k. Thus, a significant positive correlation was observed between $CO_2$ emission and water temperature. However, a negative correlation has been reported in many other studies [14,69,70]. An increasing water temperature can enhance algae growth and photosynthesis, decreasing $CO_2$ concentration and emission in the summer [71]. Although the $CO_2$ concentrations in the summer were lower than those in the spring, the $CO_2$ emissions are still the highest in the summer due to the larger value of k, similar to the findings of Tang et al. (2021) [48]. $CO_2$ is produced by aerobic respiration with the consumption of DO, promoting the development of an anaerobic environment [72]. Furthermore, an anaerobic environment accelerates the degradation of complex organic compounds to soluble small molecular organic compounds, increasing DOC concentration and further promoting $CO_2$ production. Therefore, a significantly negative correlation was observed between DO and $CO_2$ emission, in line with previous studies [48,73]. The high production of $CO_2$ by respiration could produce $H^+$ and $HCO_3^-$, decreasing pH values [74]. This may explain the negative correlation between pH and $CO_2$ emission that has been reported by most relevant studies and also by this study [48].

Shallow lakes and urban rivers with high Chl-a concentrations can uptake $CO_2$ from organic carbon respiration and even from the atmosphere, supported by the high Chl-a, DO, and pH [28,34]. This may explain the negative correlation between Chl-a and $CO_2$ emission. Factors that reflect nutrients, such as TDN and EC, are important factors affecting $CO_2$ emission [75]. The strongest correlations between $CO_2$ emission and Chl-a were observed in the winter. Within this season, the correlation between $CO_2$ emission and Chl-a was only observed in the lakes, indicating that $CO_2$ uptake in the winter had the most significant effect on the reduction in $CO_2$ emission (Figure 6). Nutrients can both promote microbial growth and mineralization, and stimulate algae growth and photosynthesis [76,77]. $CO_2$ emission was negatively correlated with TDN and EC, indicating that the increase in nutrients had a greater promoting effect on microbial respiration than algae photosynthesis. This is supported by the positive correlation between $CO_2$ emission and 16S. According to these correlations between $CO_2$ emission and environmental factors, seasonal and spatial variation can be explained. Seasonally, the highest $CO_2$ emissions were observed in the summer, due to the intense respiration with low DO and high T. The lowest $CO_2$ emissions were observed in the winter, due to the inhibition of respiration by high DO and low T [78]. Spatially, the highest $CO_2$ emissions were observed in the UR, due to low DO and high nutrient levels. The most depleted $\delta^{13}C$ in DIC was observed in the summer, and the UR verified this intense respiration. The moderate $CO_2$ emissions were observed in the LUR, due to the high DO and moderate nutrient levels. The moderate values of $\delta^{13}C$-DIC verified the moderate respiration, compare with the UR and lakes. The lowest $CO_2$ emissions were observed in the lakes, due to the high DO and low nutrient levels.

*4.4. Control Factors on $CH_4$ Emissions*

Similarly to $CO_2$ emission, seasonal variation of $CH_4$ emission was also controlled by DO and T, due to methanogens being strongly dependent on T and DO [79,80]. $CH_4$ concentrations were the highest in the spring, but $CH_4$ emissions were the highest in the summer due to the highest T in the summer, in line with the study in the Chongqing

River [9]. However, some studies reported exceptionally higher $CH_4$ emissions in winter due to the $CH_4$ accumulation under ice [75]. The lower DO reflected the intense anaerobic methanogens in the spring and summer, resulting in higher $CH_4$ emissions. Spatial variations of $CH_4$ emission were influenced by DO, pH, EC, TDN, Chl-a, and 16S. Previous studies showed that methanogens are sensitive to pH, and it is most beneficial within the range of 6 to 8 [27,70]. The $CH_4$ production would decrease outside this pH range decreases. pH values ranged from 6.9 to 10.2; hence, $CH_4$ emission exhibited a negative correlation with pH, and the higher $CH_4$ emission was observed in the UR with the lower values of pH. Nutrient levels also influence the production of $CH_4$. Alshboul et al. reported that the concentration of $CH_4$ in the drainage area with high nutrient levels was 40 times that of the level upstream [81]. Positive correlations were found between $CH_4$ emission and EC and TDN. Previous studies reported that competition for DO exists between nitrification of $NH_4^+$ and $CH_4$ oxidation, accumulating $CH_4$ in rivers and lakes [82]. The positive correlation between $CH_4$ emission and $NH_4^+$-N confirmed this speculation. Hence, the highest $CH_4$ emissions were found in the UR, also due to the higher nutrient level. However, although the TDN and EC in the LUR were higher than those in the lakes, its $CH_4$ emissions were not higher than those of the lakes, suggesting that the $CH_4$ emissions in the LUR were controlled by DO. Algae photosynthesis releases $O_2$, inhibits methanogenesis, promotes methane oxidation, and reduces the $CH_4$ concentration in water bodies [28,83]. Hence, significant negative correlations were found between $CH_4$ emission and Chl-a.

## 5. Conclusions

We investigated $CO_2$ and $CH_4$ emissions and environmental factors in the urban interconnected river–lake system in March 2021, July 2021, October 2021, and December 2021. This study presented the seasonal and spatial variations in the dissolved concentrations and emissions of $CO_2$ and $CH_4$. Our results indicated that the urban interconnected river–lake system is a hotspot of $CO_2$ and $CH_4$ emission, and the connection of river and lake decreases $CO_2$ and $CH_4$ emission of the urban rivers substantially. The $CO_2$ emissions of LUR and UR are 543.49 and 991.56 mmol $m^{-2}$ $d^{-1}$, and their $CH_4$ emissions are 1.26 and 14.82 mmol $m^{-2}$ $d^{-1}$, respectively. We found that in situ aerobic and anaerobic respiration and sewage were the sources of $CO_2$ and $CH_4$ emissions in this river–lake system. The abundant nutrients promoted microbial growth, consuming DO and thus resulting in high emissions of $CO_2$ and $CH_4$ in the UR. The nutrients also enhanced the algae growth, taking $CO_2$, resulting in the lowest emission of $CO_2$ and $CH_4$ in the lakes. The $CO_2$ emission from the lake is 365.14 mmol $m^{-2}$ $d^{-1}$, and the $CH_4$ emission is 1.25 mmol $m^{-2}$ $d^{-1}$. The moderate nutrients wreaked in situ respiration, exhibiting moderate $CO_2$ emission in the LUR. Good hydrology connectivity made short hydrology residence time, inhibiting the growth of algae that decreased the $CO_2$ uptake in the LUR. The DO concentration of LUR is 1.7 times higher than that of urban rivers, because the connection of rivers and lakes increases the DO concentration of LUR, making the $CH_4$ emission of LUR lower than that of urban rivers. Correlation analysis showed that the seasonal variation of $CO_2$ and $CH_4$ emissions was controlled by DO and T. Hence, the highest emissions of $CO_2$ were observed in the spring and the lowest in the winter, and the $CO_2$ emissions in spring were 10.7 times that in winter. The highest emissions of $CH_4$ were observed in the summer, and the lowest in the winter, and the $CH_4$ emissions in summer were 6.6 times that in winter. In summary, the connection of urban rivers and lakes changes the environmental factors, thereby varying the production and emission of greenhouse gases. However, hydrological connectivity has complex spatiotemporal variations. Further research should be conducted to improve the knowledge of the correlation between hydrological connectivity and greenhouse gas emission, providing the theoretical basis for greenhouse gas emission reduction in urban rivers.

There are large uncertainties in our calculations of carbon dioxide and methane emissions. Because we use wind speed and temperature as key variables, and do not consider factors such as river velocity and channel width, further research is needed to accurately

estimate $CO_2$ and $CH_4$ emissions. In this study, only $CH_4$ emissions diffused through the water–air interface were calculated, which were not included in the boiling volume. However, during the sampling process, we found that air bubbles were constantly emerging from the sediments in heavily polluted river sections, and the amount of $CH_4$ emissions by urban rivers may be seriously underestimated. Therefore, the amount of $CH_4$ released by air bubbles should be measured in the future.

**Author Contributions:** Conceptualization, S.L. and X.L.; methodology, C.W., S.L. and X.L.; software, C.W. and Y.X.; validation, C.W.; formal analysis, C.W.; investigation, C.W., Y.X. and X.L.; data curation, C.W. and X.L.; writing—original draft preparation, C.W.; writing—review and editing, C.W. and X.L.; visualization, C.W., Y.X. and X.L.; supervision, X.L.; project administration, X.L. All authors have read and agreed to the published version of the manuscript.

**Funding:** This work was supported by the Graduate Innovation Foundation of Wuhan Institute of Technology (CX2021448).

**Data Availability Statement:** Not applicable.

**Acknowledgments:** The authors acknowledge the kind help of our work team in fieldwork and sample analysis. We also thank anonymous reviewers and editors for their insightful comments.

**Conflicts of Interest:** No conflict of interest exist in the submission of this manuscript, and the manuscript is approved by all authors for publication. I would like to declare on behalf of my co-authors that the work described was original research that has not been published previously, and is not under consideration for publication elsewhere, in whole or in part. All the authors listed have approved the manuscript that is enclosed.

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
