# Peer review of "Interconnected River–Lake Project Decreased CO2 and CH4 Emission from Urban Rivers"

_water, doi:10.3390/w15111986_

Round 1

Reviewer 1 Report

The manuscript presents" Interconnected River-Lake project decreased CO2 and CH4 2 emission from urban rivers " which is interesting. However, the manuscript, in its present form, contains several weaknesses. Appropriate revisions to the following points should be undertaken to justify recommendations for publication.

- The abstract, what was your analytical method and statistical software?

The manuscript needs to demonstrate its relevance to an international audience, and be better situated in current scholarship. There is a rich literature on the Interconnected River-Lake project decreased CO2 and CH4 2 emission from urban rivers and diverse literature. However, the specific 'scholarly' research gap or problem (whether conceptual, theoretical, or methodological) that the manuscript seeks to address is not clearly articulated.

The manuscript needs to demonstrate its relevance to an international audience, and be better situated in current scholarship. There is a rich literature on the River-Lake relationship. However, the specific 'scholarly' research gap or problem (whether conceptual, theoretical, or methodological) that the manuscript seeks to address is not clearly articulated.

- Why is there a need to study variables affecting River-Lake relationship in general - not within the study area specifically? In other words, what is the gap in international scholarly research that this paper attempts to address? What is the scholarly research question?

Some key parameters are not mentioned. The rationale on the choice of the particular set of parameters should be explained with more details. Have the authors experimented with other sets of values? What are the sensitivities of these parameters on the results?

Some assumptions are stated in various sections. Justifications should be provided on these assumptions. Evaluation on how they will affect the results should be made.

In the conclusion section, the limitations of this study, suggested improvements of this work and future directions should be highlighted. What is the research innovation(s)?

The discussion section in the present form is relatively weak and should be strengthened with more details and justifications. There are also relatively low citations in the Discussion, suggesting that the findings are not clearly articulated in the context of broader scholarship.

In the contribution section, the paper should make a theoretical contribution, fill gaps in current knowledge.

Moreover, the manuscript could be substantially improved by relying and citing more on recent literatures about environmental hazard and River-Lake relationship such as the followings:

Varvani, et al. (2019). Investigation of the relationship between sediment graph and Hydrograph of flood events (Case Study: Gharachay river tributaries, Arak, Iran). Water Resources, 46(6):883-893.    DOI: 10.1134/S0097807819060204

Khaleghi, M.R., Varvani, J. (2018). Sediment Rating Curve Parameters Relationship with Watershed Characteristics in the Semiarid River Watersheds. Arabian Journal for Science and Engineering, 43(7), 3725–3737.

Best regards

Author Response

尊敬的编辑和审稿人:

On behalf of all the contributing authors, I would like to express our sincere appreciation for reviewers’ constructive comments concerning our article entitled “Interconnected River-Lake project decreased CO2 and CH4 emission from urban rivers”. These comments are all valuable and helpful for improving our article. According to the reviewers’ comments, we have made extensive modifications to our manuscript and supplemented extra data to make our results convincing. In this revised version, changes to our manuscript were all highlighted within the document by using red-colored text. Point-by-point responses to the two nice reviewers are listed below this letter.

Response to the Reviewer’s Comments:

Reviewer #1

Question 1: The abstract, what was your analytical method and statistical software?

Response 1: We used IBM SPSS 26.0 software to test the temporal and spatial differences of each data using the nonparametric Kruskal-Wallis method (K-W test). Using R 4.1.0 software, Spearman correlation analysis method was used to analyze the correlation between various environmental factors and CO2 and CH4 emissions. Finally, the Origin 2021 software was used to draw the boxplots of the concentration and emission of CO2 and CH4, the temporal and spatial distribution of δ13C-DIC values, and the correlation between CO2 emission and Chl-a and δ13C-DIC in each temporal and spatial. The methods and formulas used in the specific experimental methods and analytical calculations are described in detail in Sections 2.2 and 2.3.

    As suggested by reviewer #2, we have changed the title of Section 2.2 to "Experimental samples and analysis methods", changed the title of Section 2.3 to "Experimental methodology", and merged Section 2.4 into Section 2.3.

Question 2: The manuscript needs to demonstrate its relevance to an international audience, and be better situated in current scholarship. There is a rich literature on the Interconnected River-Lake project decreased CO2 and CH4 emission from urban rivers and diverse literature. However, the specific 'scholarly' research gap or problem (whether conceptual, theoretical, or methodological) that the manuscript seeks to address is not clearly articulated.

Why is there a need to study variables affecting River-Lake relationship in general - not within the study area specifically? In other words, what is the gap in international scholarly research that this paper attempts to address? What is the scholarly research question?

Response 2: This research is to improve China's understanding of river-lake connection greenhouse gas emissions, and to provide a theoretical basis for the reduction of greenhouse gas emissions in Chinese urban rivers. Because, in China, lakes and rivers are traditionally studied as independent greenhouse gas emission systems, and there is still a gap in the understanding of the temporal and spatial changes of greenhouse gas emissions in urban interconnected river-lake systems. In 2021, academician Liu Changming of the Chinese Academy of Sciences published an article on the key scientific issues and research directions of river and lake water system connectivity in the Chinese Journal of Geography. The spatiotemporal evolution mechanism of connected water pollution in river and lake systems is one of the issues that urgently need to be studied. (Liu Changming, Li Zongli, Wang Zhonggen, Hao Xiuping, Zhao Changsen. Key scientific issues and research directions of river and lake water system connectivity. Chinese Journal of Geography, 2021, 76(3):8.)

问题3:一些关键参数没有提到。应更详细地解释选择特定参数集的基本原理。作者是否尝试过其他值集?这些参数对结果的敏感性如何?

回应 3:在整个研究过程中,我们测量了 14 个环境参数,包括 ORP、DO、pH、T、EC、Chl-a、TDN、TN、DOC、TOC、硝酸盐氮、亚硝酸盐氮、氨氮和 16S 丰度,以及期间数据分析过程,通过相关性分析,对同类碳氮环境因子选择相关性最强的作为主要代表,最终筛选出对CO 2 和CH 4 排放影响最大的8环境因子出去。其他对CO 2和CH 4不敏感的环境因素排放忽略不计,只对八个环境因素中最重要的控制因素详细分析其影响排放的原理。限于篇幅,其他次要因素仅作简要分析。根据3.5节图5判断主要影响因子和次要影响因子。

问题 4:一些假设在各个部分中都有陈述。应根据这些假设提供理由。应评估它们将如何影响结果。

Response 4: Based on your suggestion, we have analyzed the reasons for the statements and assumptions in the Results section and added them at the end of the paragraph. The reasons for each hypothesis are fully dissected in the Discussion section. Section 3.1, added on page 6, lines 211 to 217. Section 3.2, added on page 7, lines 242 to 246. Section 3.3, added on page 8, lines 263 to 266. Section 3.4, added on page 8, lines 277 to 280. Section 3.5, added on page 9 to 10, lines 294 to 298, 310 to 315, and 322 to 328.

Question 5: In the conclusion section, the limitations of this study, suggested improvements of this work and future directions should be highlighted. What is the research innovation(s)?

Response 5: At the end of the conclusion, we add the limitations and future directions of this study. The innovation of our research is that there is still a big gap in China's research on greenhouse gas emissions in the system of connected urban rivers and lakes, and our research can contribute experience to make up for the gap. The conclusion section, Research limitations and future directions, was added on page 14, lines 489 to 497.

Question 6: The discussion section in the present form is relatively weak and should be strengthened with more details and justifications. There are also relatively low citations in the Discussion, suggesting that the findings are not clearly articulated in the context of broader scholarship.

Response 6: As suggested by the reviewer. We have enhanced the Discussion with some details and rationale. we have added more references to support this discussion. In section 4.1, the new citation is as follows: DO concentrations were much higher in the LUR than those in the UR, which is consistent with previous studies (Wang et al. 2022, Tang et al. 2021, Cai et al. 2023). These references are added on page 11, line 339.

In section 4.2, the new citation is as follows: the contribution of external biodegradation sourced CO2 is limited in urban rivers, while the in-situ biodegradation of organic carbon is likely more relevant to aqueous CO2 (Begum et al. 2023). This reference was added on page 11, line 366.

In section 4.3, a new reference is added: CO2 is produced by aerobic respiration with consumption of DO, promoting the development of an anaerobic environment (Solano et al. 2023). This reference is added on page 12, line 403. An increasing water temperature can enhance algae growth and photosynthesis, de-creasing the CO2 concentration and emission in the summer (Kryvenda et al. 2023). This reference was added on page 12, line 399. The lowest CO2 emissions were observed in the winter, due to the inhibition of respiration by high DO and low T (Tang et al. 2023). This reference was added on page 13, line 427.

In section 4.4, a new reference is added: Similarly with CO2 emission, seasonal variation of CH4 emission were also con-trolled by DO and T, due to methanogens is strongly depended on T and DO (Que et al. 2023). This reference was added on page 13, line 436. Algae photosynthesis releases O2, inhibits methanogenesis, promotes methane oxidation, and reduces the CH4 concentration in water bodies (Roth et al. 2023). This reference was added on page 13, line 457.

Question 7: In the contribution section, the paper should make a theoretical contribution, fill gaps in current knowledge.

Response 7: The purpose of our research on the interconnected river-lake system is to reveal the temporal and spatial variation characteristics of environmental factors, CO2 and CH4 emissions after urban rivers and lakes are connected, and to determine the impact of urban rivers and lakes on urban river CO2 and CH4 emissions.

    In the results section, we detailed the environmental factors, δ13C-DIC, the temporal and spatial distribution characteristics of CO2 and CH4 emissions, and the correlation between environmental factors and CO2 and CH4 emissions based on the data. The discussion part mainly discusses the reasons of these phenomena. Finally, the following conclusions are drawn: (1) The connection of river and lake decreasing CO2 and CH4 emission of the urban rivers substantially. The CO2 and CH4 emissions of urban rivers are 1.8 times and 11.8 times that of urban rivers connected to lakes, respectively. (2) The highest emissions of CO2 were observed in the spring and the lowest in the winter, and the CO2 emissions in spring were 10.7 times that in winter. The highest emissions of CH4 were observed in the summer and the lowest in the winter, and the CH4 emissions in summer were 6.6 times that in winter. (3) We found that in-situ aerobic and anaerobic respiration and sewage were the sources of CO2 and CH4 emission in this river-lake system. (4) The moderate nutrients wreaked the in-situ respiration, exhibiting moderate CO2 emission in the LUR. (5) The DO concentration of LUR is 1.7 times higher than that of urban rivers, because the connection of rivers and lakes increases the DO concentration of LUR, making the CH4 emission of LUR lower.

Question 8: Moreover, the manuscript could be substantially improved by relying and citing more on recent literatures about environmental hazard and River-Lake relationship such as the followings:

Varvani, et al. (2019). Investigation of the relationship between sediment graph and Hydrograph of flood events (Case Study: Gharachay river tributaries, Arak, Iran). Water Resources, 46(6):883-893.    DOI: 10.1134/S0097807819060204

Khaleghi, M.R., Varvani, J. (2018). Sediment Rating Curve Parameters Relationship with Watershed Characteristics in the Semiarid River Watersheds. Arabian Journal for Science and Engineering, 43(7), 3725–3737.

Response 8: We have checked the literature carefully and added more references on “the surface runoff formed by rainfall carry sediment into urban rivers, and the sediment layer continues to thicken, thus promoting CO2 and CH4 emissions” into the Introduction part in the revised manuscript (Varvani et al. 2019, Khaleghi and Varvani et al.2018). These are added on page 2, lines 71 to 73.

我们尽力改进了手稿,并在修订后的论文中做了一些红色标记的更改,这些更改不会影响论文的内容和框架。我们衷心感谢编辑/审稿人的热情工作,并希望更正能够获得批准。再次非常感谢您的意见和建议。

Reviewer 2 Report

(1) The quality of Figure 1 should be improved. The image in Figure 1 regarding the study area is somewhat blurry, and its resolution needs to be improved.

(2) I think title of section 2.2 should be changed as "Experimental Samples and Analysis Methods". If fact, this section introduces the experimental materials and methods. Besides, title of Section 2.3 should be changed as "Experimental Methodology". In fact, this section introduces discusses how to process experimental data. Additionally, it is recommended to merge sections 2.4 and 2.3. This is because making section 2.4 a separate part is a bit hollow.

(3) For investigations in manuscript, it is necessary to add moderate mechanism (cause) exploration after data analysis in each section, and data analysis alone cannot be carried out. For example, in Section 3.2, the reasons for these phenomena in Figure 2 should be appropriately explained.

(4) I see most of the R2(Correlation coefficient) in Figure 6and Figure 7 is lower than 0.50. Does this mean that the correlation is not good? If the correlation is not good, what is the significance of the research?

(5) In Abastract and Conclusions, some quantitative descriptions are needed to enhance their persuasiveness. Therefore, it is recommended to rewrite the abstract and conclusion to make them appear persuasive.

(6) The follwing references should be cited to support the statement: Inland waters, including rivers, reservoirs, and lakes, are essential to the global carbon cycle and are a significant source of atmospheric greenhouse gases. ① Geomechanics and Geophysics for Geo-Energy and Geo-Resources, 2022, 8(2): 82.; ② Environmental Science and Pollution Research, 2022, 29(51): 77737-77754.; ③ https://doi.org/10.1007/s11356-023-26279-9 (Just search for the website directly in the address bar). 

Moderate editing of English language in manuscript. There are some long and difficult sentences in the manuscript, which can easily make people confused.

Author Response

On behalf of all the contributing authors, I would like to express our sincere appreciation for reviewers’ constructive comments concerning our article entitled “Interconnected River-Lake project decreased CO2 and CH4 emission from urban rivers”. These comments are all valuable and helpful for improving our article. According to the reviewers’ comments, we have made extensive modifications to our manuscript and supplemented extra data to make our results convincing. In this revised version, changes to our manuscript were all highlighted within the document by using red-colored text. Point-by-point responses to the two nice reviewers are listed below this letter.

Response to the Reviewer’s Comments:

Reviewer #2

Question 1: The quality of Figure 1 should be improved. The image in Figure 1 regarding the study area is somewhat blurry, and its resolution needs to be improved.

Response 1: We have adjusted the resolution of Figure 1 to 300dpi, and increased the font size of the sampling point numbers.

Question 2: I think title of section 2.2 should be changed as "Experimental Samples and Analysis Methods". If fact, this section introduces the experimental materials and methods. Besides, title of Section 2.3 should be changed as "Experimental Methodology". In fact, this section introduces discusses how to process experimental data. Additionally, it is recommended to merge sections 2.4 and 2.3. This is because making section 2.4 a separate part is a bit hollow.

Response 2: We think this is an excellent suggestion. We changed the title of section 2.2 to "Experimental samples and analysis methods", changed the title of section 2.3 to "Experimental methodology", and merged section 2.3 and section 2.4. The revised content is on pages 4 and 5.

Question 3: For investigations in manuscript, it is necessary to add moderate mechanism (cause) exploration after data analysis in each section, and data analysis alone cannot be carried out. For example, in Section 3.2, the reasons for these phenomena in Figure 2 should be appropriately explained.

Response 3: According to your suggestion, we have added the reasons for each phenomenon after the data analysis in section 3.1, section 3.2, section 3.3, section 3.4 and section 3.5. The reasons for each phenomenon are fully dissected in the Discussion section. Section 3.1, added on page 6, lines 211 to 217. Section 3.2, added on page 7, lines 242 to 246. Section 3.3, added on page 8, lines 263 to 266. Section 3.4, added on page 8, lines 277 to 280. Section 3.5, added on page 9 to 10, lines 294 to 298, 310 to 315, and 322 to 328.

Question 4: I see most of the R2(Correlation coefficient) in Figure 6 and Figure 7 is lower than 0.50. Does this mean that the correlation is not good? If the correlation is not good, what is the significance of the research?

Response 4: In figure 6 and figure 7, R2 less than 0.5 means that the correlation is not good, but there are also R2 greater than 0.5. The comparison of such good correlation results and bad correlation results can help us to judge which season of the environmental factor is the main influencing factor in each season of UR, LUR and lake.

In figure 6, the R2 of the three seasons of UR is less than 0.5. Combined with figure 4, it can be shown that the CO2 emission of urban rivers is much higher than that of LUR and lakes, and CO2 mainly comes from the decomposition of organic matter, and the amount of CO2 consumed by Chl-a has little impact on emissions. The same goes for summer and fall in LUR and lakes. The same goes for summer and fall in LUR and lakes. However, the CO2 emissions of LUR and lakes were the lowest in winter, and the CO2 emissions were also negatively correlated with Chl-a, with R2 greater than 0.5. It shows that the amount of CO2 produced by the decomposition of LUR and lake organic matter decreases sharply in winter, and the CO2 consumed by Chl-a has a great impact on CO2 emissions, resulting in the higher the concentration of Chl-a, the lower the CO2 emissions.

In figure 7, CO2 emissions in each season and δ13C-DIC values are negatively correlated with R2 greater than 0.5 and P less than 0.01, indicating that the higher the CO2 concentration, the higher the proportion of CO2 produced by the decomposition of organic matter, and the same is true for LUR. However, the R2 of UR and lakes is less than 0.5, and the P value is less than 0.05. There may be other sources of influence on the δ13C-DIC values of UR and lakes.

Question 5: In Abastract and Conclusions, some quantitative descriptions are needed to enhance their persuasiveness. Therefore, it is recommended to rewrite the abstract and conclusion to make them appear persuasive.

Response 5: As suggested by the reviewers, we have supplemented the Abstract and Conclusions with quantitative data. In the abstract on page 1, we added "CO2 and CH4 emissions from urban rivers were much higher than those from the lakes, which are 2.7 times and 11.9 times that of lakes, respectively." in line 18. Added "The average CO2 and CH4 emissions of urban rivers are 991.56 and 14.82 mmol m-2 d-1, respectively." in line 22. Added "The average CO2 emission of LUR is 543.49 mmol m-2 d-1." in line 25. Line 27 added "The average CH4 emission of LUR is 1.26 mmol m-2 d-1." In Abstract lines 29 to 32 and Conclusion lines 479 and 482 rewritten, "Hence, the highest emissions of CO2 were observed in the spring and the lowest in the winter, and the CO2 emissions in spring were 10.7 times that in winter. The highest emissions of CH4 were observed in the summer and the lowest in the winter, and the CH4 emissions in summer were 6.6 times that in winter."

    Added, " The CO2 emissions of LUR and UR are 543.49 and 991.56 mmol m-2 d-1, and their CH4 emissions are 1.26 and 14.82 mmol m-2 d-1, respectively." to lines 465 and 467 on page 13 of the Conclusions. Added, " The CO2 emission from the lake is 365.14 mmol m-2 d-1 and the CH4 emission is 1.25 mmol m-2 d-1." to line 471 on page 13 of the Conclusions. Added, " The DO concentration of LUR is 1.7 times higher than that of urban rivers, because the connection of rivers and lakes increases the DO concentration of LUR, making the CH4 emission of LUR lower than that of urban rivers." to lines 475 to 478 on page 14 of the Conclusions.

Question 6: The follwing references should be cited to support the statement: Inland waters, including rivers, reservoirs, and lakes, are essential to the global carbon cycle and are a significant source of atmospheric greenhouse gases. Geomechanics and Geophysics for Geo-Energy and Geo-Resources, 2022, 8(2): 82.; ② Environmental Science and Pollution Research, 2022, 29(51): 77737-77754.; ③ https://doi.org/10.1007/s11356-023-26279-9 (Just search for the website directly in the address bar). 

Response 6: We cite your suggested literature in support of this statement: Inland waters, including rivers, reservoirs, and lakes, are essential to the global carbon cycle and are a significant source of atmospheric greenhouse gases (Li et al. 2022, Li et al. 2023, Li and Wu 2022). Located on line 46 on page 2 of the Introduction section.

We tried our best to improve the manuscript and made some changes marked in red in revised paper which will not influence the content and framework of the paper. We appreciate for Editors/Reviewers’ warm work earnestly, and hope the correction will meet with approval. Once again, thank you very much for your comments and suggestions.

Round 2

Reviewer 1 Report

I have carefully read the manuscript, paying particular attention to the responsiveness of the authors to the original reviewers' concerns. To me, the authors have done a reasonable job of addressing the primary concerns. As such, I am happy to recommend that the article be accepted.

Reviewer 2 Report

After the authors' revisions, I believe that the manuscript has met the requirements for acceptance and publication.